# Neuroferritinopathy Human-Induced Pluripotent Stem Cell-Derived Astrocytes Reveal an Active Role of Free Intracellular Iron in Astrocyte Reactivity

**DOI:** 10.3390/ijms26136197

**Published:** 2025-06-27

**Authors:** Andrea Stefano Moro, Chiara Balestrucci, Anna Cozzi, Paolo Santambrogio, Sonia Levi

**Affiliations:** 1Faculty of Medicine, Vita-Salute San Raffaele University, 20132 Milan, Italy; moro.andrea@hsr.it (A.S.M.); balestrucci.chiara@hsr.it (C.B.); 2Division of Neuroscience, IRCCS San Raffaele Scientific Institute, 20132 Milan, Italy; cozzi.anna@hsr.it (A.C.); santambrogio.paolo@hsr.it (P.S.)

**Keywords:** neuroferritinopathy, iron homeostasis, ferroptosis, neuroinflammation, neurodegenerative diseases

## Abstract

Increased iron levels, common in neurodegenerative diseases, correlate with disease severity, suggesting a role in the pathological process. Recently, efforts have been made to understand the role of iron in cerebral inflammatory processes. Employing astrocyte cell models of genetic neurodegenerative pathologies characterized by iron imbalance, such as the neurodegeneration with brain iron accumulation disorders, can provide valuable insights into astrocytes reactivity, a pivotal process in brain inflammation. Specifically, we employed human-induced pluripotent stem cell-derived astrocytes from Neuroferritinopathy, where iron accumulation is primary. After confirming iron accumulation and the deregulation of proteins involved in iron management, we observed that at 35 days since the beginning of differentiation, the elevated iron levels not only trigger ferroptosis but also place the astrocytes in a reactive state. This is evident in the higher extracellular concentrations of IL-6, IL-1β, and glutamate, along with changes in morphology, genes, and proteins involved in astrocyte reactivity. Interestingly, by day 60, IL-6 and IL-1β levels drop below those of the controls, and we observe a reversal in most of the factors considered. Moreover, at day 60, it is possible to observe not only increased senescence but also ferroptosis. These findings demonstrate that iron plays a primary role in inducing astrocyte reactivity.

## 1. Introduction

The dysregulation of iron homeostasis in the brain is a common feature across many neurodegenerative diseases, with the precise role of iron in the progression of neurodegeneration still under debate [1]. While the role of neuroinflammation in neurodegenerative processes is becoming increasingly well defined, the involvement of iron in neuroinflammation remains largely mysterious. Disease models of neurodegeneration with brain iron accumulation (NBIA) provide an opportunity to investigate the specific role of iron in astrocytes reactivity and in its role in contributing to neurodegeneration [2].

Neuroferritinopathy (NF, OMIM# 606159, or NBIA3) [3] belongs to the family of NBIA disorders [2], which are genetic conditions characterized by abnormal iron deposition in the brain [1]. NF is inherited as an autosomal dominant trait, with 100% penetrance, and typically manifests between the ages of twenty and fifty. The disorder is caused by mutations in the gene encoding the L chain of ferritin (*FTL1*), resulting in a range of extrapyramidal symptoms and the full spectrum of signs and symptoms characteristic of movement disorders, alongside progressive cognitive decline as the disease advances. NF is an extremely rare condition, and consequently, data on its prevalence within the general population is currently unavailable. It affects both males and females equally, although some families have reported a male predominance. The symptomatic triad of oro-mandibular dyskinesia (65%), dysarthrophonia, impaired voice and language, and action-specific facial dystonia (63%) is a hallmark feature of NF. Iron blood chemistry profiles are usually normal, except for serum ferritin levels, which are often lower than normal, although some cases may initially present with normal serum ferritin values [4,5].

Early MRI findings in the symptomatic phase reveal hypodensity on T2* and SWI at the level of the dentate nucleus, red nucleus, substantia nigra, putamen, globus pallidus, thalamus, caudate nucleus, and motor cortex [6,7]. As the disease progresses, T2-weighted images show hyperintensity in the basal ganglia due to edema and gliosis preceding degeneration [7], and in advanced stages, cystic cavitation areas develop [8]. The “eye of the tiger” sign has also been observed in three patients with NF [9].

The disease was first identified in a family from northern England by Curtis and colleagues [3], who discovered that the insertion of an adenine at position c460 of exon 4 of the *FTL1* gene was the causative factor of this neurodegenerative disorder. Initially, it was hypothesized that the disease was limited to the Cumbria region of northern England, originating from a common ancestor to all identified cases. However, subsequent research has identified eleven types [3,10,11,12,13,14,15,16,17,18] of mutations across different ethnic groups, each with distinct phenotypic expressions and varying ages of onset, linked to the specific mutations.

The *FTL1* gene is located on chromosome 19q13.33 and comprises 4 exons and 3 introns. The encoded peptide combines with H-ferritin chains, codified by *FTH*, to form the 24-mer heteropolymer ferritin (Ft), which serves as the primary iron storage protein. All mutations associated with NF, except for one, are located within a short 58-nucleotide DNA fragment in exon 4 of the gene [19]. These mutations involve single or multiple nucleotide changes, ranging from 2 to 16 bases, leading to alterations in the sequence and length of the C-terminal portion of the encoded protein. Notably, the *FTL1* c.469_484dup mutation (See Material and Methods), involving the duplication of 16 nucleotides, has been identified in two patients from distinct and geographically distant ethnic backgrounds, namely Japan [7] and Italy [20]. Recent findings suggest that similar structural alterations in ferritin can also arise from mutations in the H-ferritin chain, indicating that the *FTH* gene may also contribute to NF. The truncated 79-ammino-acid resulting protein led to a novel pediatric NF [21].

A common characteristic across the different variants of NF is the structural alteration of the terminal portion of the ferritin subunit, specifically the E-helix. In the fully assembled 24-mer ferritin, this E-helix plays a crucial role in forming a hydrophobic channel along the four-fold axes of symmetry [22]. The incorporation of these altered subunits into ferritin heteropolymers disrupts the proper formation of this channel, leading to increased pore sizes, which in turn compromises the protein’s ability to effectively retain deposited iron [19,23]. The resulting increase in cytosolic redox-active iron triggers the upregulation of iron-dependent ferritin translation. This continuous production of faulty peptides further promotes the aggregation of ferritin and iron within the cytosol [11].

Indeed, NF is characterized by histopathological lesions associated with the presence of ferritin and iron aggregates in both nervous and other tissues. Macroscopic examination of the brains of NF patients has revealed the presence of spherical granules positive for iron and ferritin in the globus pallidus, forebrain, and cerebellum [3,14,17,24,25]. These aggregates are predominantly located extracellularly, but intranuclear and intracytoplasmic aggregates have also been observed in neurons, microglia, and oligodendrocytes, containing iron and ferritin composed by both wild-type chains, ferritin light chain (FtL) and ferritin heavy chain (FtH), and by the pathological variant FtL [3,14,17,25].

Studies conducted on cellular disease models have provided evidence that the alteration of ferritin function leads to an increase in cytosolic redox-active iron, triggering a cascade of events that ultimately result in ferritin aggregation and impairment of both proteasomal and lysosomal systems [26]. Ultrastructural analysis of NF transgenic mouse brains has further confirmed the presence of iron–ferritin complexes accompanied by signs of oxidative damage and revealed impairment of the lysosomal compartment with lipofuscin formation [27]. Moreover, additional results have been obtained through studies on NF fibroblasts [11,28], neural progenitor cells (NPCs), and induced pluripotent stem cell (iPSC)-derived glutamatergic neurons [11]. These neurons exhibited a phenotype similar to fibroblasts, characterized by reduced ferritin functionality, cytosolic aggregates, altered iron homeostasis parameters, and increased oxidative stress and senescence, as previously reported [11]. Key findings from this neuronal model include the spontaneous onset of ferroptosis in iPSC-derived neurons, which could be suppressed by ferrostatin-1 and iron chelation. Furthermore, evidence suggests that ferroptotic cell death may occur following the establishment of a senescent phenotype in neurons [11].

Another factor that can contribute to neurodegeneration is neuroinflammation—a systemic response involving various cell types. Microglia, for example, are the resident macrophages of the central nervous system and play a key role in immune surveillance and inflammatory responses [29]. Their involvement has been partially studied in neurodegenerative diseases such as Parkinson’s disease (PD) and Alzheimer’s disease (AD) [30,31]. However, our experiments have shown that iron metabolism is particularly disrupted in astrocytes [32,33,34], and that these cells can exist in different functional states, ranging from neuroprotective to pro-inflammatory. One such pro-inflammatory state is reactive astrogliosis, which has been associated with disease progression [35]. Although astrocyte reactivity is a core component of neuroinflammation, the concurrent involvement of multiple cell types complicates the analysis of this phenomenon.

To address this, we employed a simplified astrocytic model: d-astrocytes derived from iPSCs of NF patients. This model enables the study of astrocyte-specific responses in the context of elevated intracellular free iron. Using this system, we observed that iron-loaded d-astrocytes express markers of reactivity during the early days following differentiation, suggesting the acquisition of a reactive phenotype. However, this behavior diminishes at later stages in vitro, likely due to the onset of cellular senescence and ferroptosis.

## 2. Results

### 2.1. Development of hiPS-Derived Astrocytes from NF Patients and Controls

hiPS-derived astrocytes (d-astrocytes) were differentiated from previously obtained hiPS clones [11] of one healthy subject (control), one isogenic control, obtained by CRISPR/Cas9 technique (R-NF1#44), and two NF patients: one carrying the *FTL1* 469_484dup mutation, named NF1#1 [20], and the other carrying the *FTL1* 351delG_InsTTT mutation (better described in Materials and Methods), named NF2#11 [11]. NF patients and control hiPS clones were differentiated in d-astrocytes as previously reported [36] and schematized in Appendix A. Approximately more than 90% of cells were positive for the specific markers Glial fibrillary acidic protein (GFAP) and Excitatory amino acid transporter 2 (EAAT2), demonstrating that this culture was almost exclusively composed of astrocytes (Appendix A).

### 2.2. NF d-Astrocyte Is a Model of Iron Dys-Homeostasis

To determine whether the NF d-astrocyte model displayed the expected iron dysregulation, we quantitatively analyzed key proteins regulated by the iron-responsive element (IRE) machinery—FtH, ferroportin (Fpn), divalent metal transporter 1 (DMT1), and transferrin receptor 1 (TfR1) (iron storage, iron exporter- and iron importer-proteins, respectively). The IRE pathway is central to maintaining intracellular iron homeostasis, and changes in these proteins can reveal underlying defects in iron regulation.

As shown in the upper panels of Figure 1A, FtH and Fpn levels were significantly higher in patient astrocytes, suggesting an increased intracellular iron load and a compensatory attempt to export excess iron. Conversely, DMT1 and TfR1 (lower panels in Figure 1A) were significantly lower in patient cells, indicating reduced iron uptake. These molecular alterations are fully consistent with the pathology of NF, where dysregulated iron metabolism is a defining feature.

Iron accumulation was further confirmed by Perls staining, which revealed iron deposits that were already visible after 35 days within patient astrocytes and that constantly increased up to 60 days (Figure 1B). Together, these findings underscore the validity of this NF astrocyte model for investigating disease-related mechanisms of iron dys-homeostasis.

### 2.3. NF d-Astrocytes Show Reactive Phenotype

Morphological alterations were observed during the maintenance of the differentiated cell line, which exhibited different shapes at increasing time points. We chose the two time points (at day 35 and 60) where the morphological differences were more evident. The NF cells displayed a more stellated and elongated profile on day 35 with respect to the control cells; this pattern reversed on day 60, along with an overall increase in cell area (Figure 2A). These thin and stretched shapes are associated with astrocytic reactive phenotypes. Recognizing that days 35 and 60 might be ideal for studying astrocyte reactivity, we examined the levels of extracellular molecules typically associated with a reactive phenotype. By measuring extracellular IL-6 levels, we observed that in both patients, IL-6 increased until around day 35 and then abruptly dropped to levels even lower than those of the controls (Figure 2B). We found that IL-1β and IL-6 behaved similarly: both had higher levels on day 35 and then became lower than control levels on day 60. Indeed, linear mixed-effect analysis (see Materials and Methods) showed that the fixed effects of the pathology and their interactions with time were significant for IL-6 and IL-1β. Glutamate, however, was higher only in patient cells on day 35 and returned to control levels by day 60, with only the interaction effect being significant for glutamate. Overall, these findings reveal opposing trends in patients between days 35 and 60 (Figure 3A).

To further determine whether these data reflect astrocytic reactivity, we analyzed the behavior of key proteins and genes involved in astrocyte processes by Western blot (WB). Nuclear Factor Erythroid 2-related factor 2 (NRF2) expression levels were lower in patient cells on day 35 but recovered by day 60. In contrast, Lipocalin 2 (LCN2) increased on day 35 and returned to levels comparable to controls by day 60. Regarding EAAT2, no significant effects were observed, suggesting that differentiation issues were unlikely (Figure 3B).

Finally, qRT-PCR results supported the presence of astrocytic reactivity in patient cells on day 35, which was no longer evident on day 60 (Figure 4). Significant effects were observed for IL-6 and iNOS, both of which were higher in patient cells on day 35 but fell below control levels by day 60. For GFAP, the main difference emerged on day 60, where patient cells expressed substantially lower levels; indeed, only the interaction was significant. Conversely, TNFα was not more highly expressed in patients on day 35, as its interaction term was not significant.

### 2.4. NF d-Astrocytes Showed Iron Alteration, Senescence, and Ferroptosis

To assess whether the observed changes in astrocytic state were driven by alterations in iron homeostasis, we measured total iron content, the cytosolic and mitochondrial labile iron pools (LIP), verified the cellular senescence status, and evaluated the key ferroptosis markers, including malondialdehyde (MDA) and Glutathione peroxidase 4 (GPX4).

In control cells, neither total iron nor cytosolic LIP levels changed significantly between day 35 and day 60. In contrast, patient cells showed elevated levels of LIP on day 35, which further increased by day 60 (Figure 5). Interestingly, no significant changes were observed for the mitochondrial LIP (Appendix A), suggesting that mitochondrial iron dysregulation plays a lesser role in this context. In fact, we did not detect any alterations in heme, unlike those observed in another form of NBIA, Pantothenate Kinase-Associated Neurodegeneration [32] (Appendix A).

Our previous data on NF-glutamatergic neurons demonstrated that iron overload triggered senescence and ferroptosis [11]. Thus, we evaluate astrocyte senescence, measuring β-Galactosidase activity. As depicted in Figure 6A, senescence in d-astrocytes increased significantly from day 35 to day 60. On day 35, patient cells already showed elevated senescence levels compared to controls, and these levels rose dramatically by day 60 (Figure 6A). To further substantiate these findings—especially the marked differences observed at day 60—we measured p62 levels. The results confirmed that p62 expression was significantly higher in patient samples, supporting the notion of enhanced astrocyte senescence in this model (Figure 6B).

Furthermore, the increase in total iron and cytosolic LIP is normally paralleled by changes in markers of ferroptosis. Indeed, MDA levels—a marker of lipid peroxidation—were higher in patient cells on day 35 and increased further on day 60. Conversely, GPX4 levels, which help protect cells from ferroptosis by detoxifying lipid peroxides, were lower in patient cells on day 35 and decreased even further by day 60 (Figure 7).

Overall, these findings indicate that patient astrocytes not only accumulate more iron over time but also exhibit enhanced markers of ferroptosis, suggesting a link between iron dysregulation and ferroptotic cell damage in this model.

## 3. Discussion

Although the contribution of neuroinflammation to neurodegenerative processes is becoming more clearly understood, the role of iron is still largely unknown. It is difficult to study role of iron in neuroinflammation [37] because artificially altering intracellular iron concentrations is challenging due to numerous homeostatic mechanisms (IRP-system) that meticulously regulate iron levels [38]. For instance, direct injection of iron into the cerebral parenchyma has been shown to cause neuronal death [39]; however, this method is not only unphysiological but also reveals little about the underlying intracellular processes. Likewise, iron-loading experiments via the pyrithione [40], which affect the cellular membrane, can create artifacts that are not readily translatable. For these reasons, we decided to use pathological models in which brain iron accumulation occurs naturally, as in NF.

We used d-astrocytes from patients with NF to evaluate the role of cytosolic iron overload in determining astrocyte reactive phenotypes.

Astrocytes are cells widely distributed in the CNS and have various functions [41]. A recent study has clarified the role of astrocytes in orchestrating the transfer of iron from the bloodstream to brain tissue [42]. Through both in vivo and in vitro experiments, the authors demonstrated that astrocytes react to changes in intracellular iron levels by releasing Hepcidin (Hep), the systemic iron-hormone. They respond to elevated intracellular iron levels by increasing the secretion of Hep, which modulates the total level of iron in the brain by interacting with Fpn at the blood–brain barrier [42]. Astrocytes thus exert a significant influence in regulating the iron content in brain tissue. Furthermore, they show significant adaptability, as they undergo changes in structure and function when pathological processes affecting the CNS are triggered, such as neurodegeneration. Following these alterations, resting astrocytes undergo polarization to become reactive astrocytes, which exhibit beneficial or detrimental effects depending on their specific reactivity profile. Indeed, transcriptomic analysis of reactive astrocytes has delineated distinct phenotypes which can be neurotoxic or neuroprotective [43].

D-astrocytes from NF patients were obtained and characterized, and an investigation of cellular iron metabolism provided compelling evidence of iron accumulation within these cells.

NF-derived astrocytes displayed an accumulation of iron aggregates, notably prominent within their cytoplasm, as evidenced by positive specific Perls staining. In NF d-astrocytes, cytosolic iron accumulation was corroborated by the variable expression of iron-dependent proteins: biochemical analysis revealed heightened levels of FtH, and Fpn. Conversely, there was a downregulation of TfR1, which captures iron from Tf in the bloodstream and transports it into cells, as well as DMT1, responsible for facilitating iron uptake into cells and its cellular acquisition. The abovementioned iron-dependent proteins are regulated by the IRP-IRE machinery, which detects an excess of redox-active iron in the cytosol and triggers post-transcriptional regulation of iron-related protein expression, ensuring an appropriate cellular response to iron overload. The observation of these astrocytes over time has highlighted a morphological change typical of reactive astrocytes accompanied by the secretion of interleukins and expression of astrocytic reactivity markers that correspond to the dysregulation of iron homeostasis. The increase in intracellular iron accumulation over time establishes a cascade of events that disrupts the oxidative balance within cells, leading to ROS generation that inflicts damage upon cellular constituents, thus impeding normal cellular functions. The oxidative stress triggered by iron overload elicits a series of adverse effects on cellular physiology, leading to senescence [11] and preparing cells for ferroptosis, a regulated form of cell death reliant on iron. Ferroptosis is characterized by the accumulation of lipid peroxides, protein oxidation, and DNA damage [44]. Notably, in NF d-astrocytes, there were increased levels of MDA, a well-recognized indicator of lipid peroxidation. Furthermore, the manifestation of the ferroptosis phenotype necessitates the inactivation of GPX4, an antioxidant enzyme dependent on glutathione for its activity. The expression levels of GPX4 were diminished in NF d-astrocytes, indicating a potential mechanism contributing to the susceptibility of these cells to ferroptotic death.

In this study, the use of d-astrocytes derived from NF provided crucial insights into the role of iron in astrocyte reactivity. Our data suggest that an increase in redox-active intracellular iron alone is sufficient to initiate astrocyte reactivity; however, prolonged iron accumulation eventually leads to senescence. The correlation between iron and senescence is well established and was also observed in d-astrocytes from another NBIA [33]. 

The results obtained in these experiments further support the hypothesis that senescence acts as a compensatory strategy in response to iron accumulation. Nevertheless, using NF—where the impaired pathway directly involves iron—offers a more reliable perspective on the ability of intracellular iron to induce a reactive phenotype. An alternative explanation might be that iron triggers senescence first, and that senescent cells then release chemical signals to activate astrocyte reactivity. However, our observation of a non-secretory senescent phenotype, characterized by lower IL-6 and IL-1β levels compared to controls, makes this second explanation less likely.

Collectively, the results obtained from the characterization of NF d-astrocytes suggested that the utilization of this model facilitates the recapitulation of the peculiar characteristics of the pathological phenotype, and highlighted a primary role of redox-active iron in trigging inflammation. 

From a therapeutic perspective, our results suggest that the use of blood–brain barrier-permeable chelators may modulate the astrocytic cellular state by influencing its reactivity. Although, the use of iron chelators in neurodegenerative diseases should be carefully dosed to avoid the detrimental effects previously observed in clinical trials of PD [45,46] and AD [47].

## 4. Materials and Methods

### 4.1. Cases

Cases of patients carrying the *FTL1* 469_484dup (p.Leu162Argfs*24) heterozygous mutation (named NF1) and the heterozygous nucleotide insertion 351delG_InsTTT (p.Gly118Leufs*19) in *FTL1* (named NF2) have been described previously [11,20]. The isogenic control, named R-NF1, was obtained by performing a gene knock-in of the *FTL1* WT allele in the NF1 iPSCs by combining CRISPR/Cas9 with a single-stranded oligodeoxynucleotide donor template and a sgRNA, specifically directed to the 469_484dup mutation [11]

### 4.2. Cell Models

Independent hiPS clones for one normal subject (control) and one isogenic control R-NF1 have been previously generated and described [3]. The same hiPS clones for controls were differentiated into a pure and stable population of mature astrocytes as in [33]. Briefly, NPCs were seeded onto iMatrix-511 (Ambio, Clearwater, SC, USA)-coated plates and grown in DMEM-F12 (Euroclone, Pero, Italy), supplemented with 1% Pen/Strep (Sigma, St. Louis, MI, USA), 2 mM L-glutamine (Sigma), N2 (1:100; Life Technologies, Carlsbad, CA, USA), B27 (1:200; Life Technologies), and bFGF (20 ng/mL; Tebu-Bio, Perray en Yvelines, France). When the cell culture reached a confluence of approximately 60%, the medium was supplemented with 20% FBS (Euroclone), and the medium was changed every 2–3 days. The culture was maintained for more than 30 days to produce astrocytes with a good level of maturation.

### 4.3. H-Ferritin Quantification

The cell extracts were analyzed for ferritin by ELISA assay based on monoclonal antibodies specific for the human FtH, named rH02 (homemade), calibrated on the corresponding recombinant homopolymer [48]. The absorbance was read at 490 nm with a micro-plate reader.

### 4.4. Perls Staining

hiPSC d-astrocytes from NF patients and controls, grown on coverslips for 72 h, were fixed in 4% paraformaldehyde and were stained for iron content via the Perls reaction incubating them for 1 h in 1% potassium ferrocyanide (K_4_[Fe (CN)_6_] ∙ 3H_2_O) and 1% hydrochloric acid in distillated water. Cells were counterstained with Nuclear Fast Red (Sigma). Images were taken on a Zeiss AxioImager M2m equipped with AxioCam MRc5 (AxioVision version 4.9), using a 40× objective.

### 4.5. Immunoblotting

Pellets of hiPSC-derived astrocytes from patients and controls were first lysed in RIPA lysis buffer (10 mM Tris-HCl pH 7.4; 150 mM NaCl; 1% Triton X-100, 0.1% SDS; 0.1% sodium deoxycholate and protease inhibitor cocktail). After centrifugation the total proteins of the supernatant were quantified using BCA Protein assay (Pierce) calibrated with Bovine Serum Albumin. Immunoblotting was performed after the separation of soluble proteins (25 μg) by SDS-PAGE as in [49]. Specific antibodies and conditions are listed in Appendix A. The signal was revealed using an ECL Prime Kit (Amersham) and detected with a ChemiDoc MP Imaging System (BIORAD).

### 4.6. Determination of LIP

Cytosolic and mitochondrial LIP were measured in d-astrocytes incubated for 24 h in 96-well plate, using the iron-sensitive fluorescent probe calcein (Thermo Fisher Scientific, Waltham, MA, USA) and rhodamine B-[(1,10-phenanthrolin-5-yl)-aminocarbonyl] benzyl ester (RPA) (Squarix Biotechnology, Marl, Germany), respectively, as described in [32]. Basal fluorescence was acquired using Arrayscan XTI HCA Reader (Thermo Fischer Scientific) equipped with an LD Plan-NEOFLUAR 20×/0.4 NA objective (Zeiss, Jena, Germany). Excitation LED (386/23 nm) was used for Hoechst, LED (485/20 nm) was used for calcein, and LED (549/15 nm) was used for RPA, with a pentaband BGRFRN dichroic mirror and emission filter used for both. Total calcein or RPA fluorescence was measured after the release of iron bound to the probes by the addition of the specific iron chelator PIH (final concentration 1 mM for calcein and 2 mM for RPA) for 30 min. The difference between total and basal fluorescence represents the cytosolic or mitochondrial LIP. The results were normalized using Hoechst fluorescence as an estimation of the number of cells.

### 4.7. SA-β-Gal Activity Detection

The cells were analyzed according to the protocol “Senescence Cells Histochemical Staining” (Sigma) [11]. In brief, the cells were fixed with a solution containing 20% formaldehyde and 2% glutaraldehyde, stained with a solution of X-gal, and incubated at 37 °C for 18 h. The cells were observed under a microscope with bright light; the blue stained cells and total number of cells were counted, and the percentage of cells expressing β-galactosidase was calculated

### 4.8. Heme Content Determination

Pellets of d-astrocytes from NF patients and controls were treated with 0.5 mL with 98% formic acid for 15 min at room temperature.

The absorbance of the clarified supernatant for each sample was measured at a wavelength of 400 nm. Then, heme concentration had been determined using the molar extinction coefficient of 1.56 × 105 M^−1^cm^−1^.

The obtained values have been normalized on the total proteins of each sample using the commercial kit Bio-Rad Protein Assay (BioRad).

### 4.9. Measurement of Released Glutamate

Cells were grown in astrocyte medium supplemented with 20% dialyzed serum. Astrocytes were incubated in medium without serum for 1 h prior to the glutamate test. The amount of glutamate secreted in the medium was determined using the Glutamate Assay Kit Colorimetric (Abcam, Cambridge, UK). Astrocytes were then washed with PBS and lysed in 20 mM Tris-HCl. pH 7.4 and 1% Triton X-100. The total protein content was measured as above in clear supernatants and used to normalize the amount of glutamate.

### 4.10. qRT-PCR Analysis

Total RNA was extracted using the RNeasy Mini Kit (QIAGEN, Hilden, Germany). Reverse transcription was performed with the High-Capacity cDNA Reverse Transcription Kit (Thermo Fisher Scientific). Quantitative real-time PCR was conducted using the TaqMan Gene Expression Master Mix (Thermo Fisher Scientific) with the following TaqMan Gene Expression Assays: IL-6 (Hs00174131_m1), iNOS (Hs01075521_m1), GFAP (Hs00909233_m1), TNFα (Hs00174128_m1), and beta-actin (Hs04023880_g1) as the reference gene. The reactions were performed on a CFX96 Touch Real-Time PCR Detection System (Bio-Rad) under the following thermal cycling conditions: initial denaturation at 95 °C for 10 min, followed by 40 cycles of denaturation at 95 °C for 15 s and annealing/extension at 60 °C for 1 min. Cycle threshold (Ct) values were calculated using the baseline-threshold method.

### 4.11. Morphological Analysis

After incubation with antibodies (anti β-actin and 2nd antibody), cells were washed with PBS at 4 °C supplemented with 1% BSA, rinsed with cold PBS, fixed with 4% paraformaldehyde, and analyzed via retrospective fluorescence analysis. Fluorescence images were acquired using a LEICA TCS SP5 laser scanning confocal microscope (Wetzlar, Germany), equipped with appropriate excitation/emission filters. Laser parameters were set to avoid pixel saturation, and images were captured at a resolution of 1024 × 1024 pixels, with a 12-bit depth, using a 40×/1.25 NA oil-immersion objective. Using a custom algorithm, the images were binarized, and key morphological features (Major Axis, Minor Axis, and Area) were extracted. The script was developed in MATLAB R2025a (Mathworks, Natick, MA, USA) using standard built-in functions/plugins, as previously described.

### 4.12. Human Interleukin ELISA Assay

The ELISA assays on human IL-6 and IL-1*β* are performed on d-astrocytes growth medium of patients and controls using ELISA FLEX kit (HRP) for quantitative determination of native and recombinant human IL-6 and IL1*β* (Product Code: 3460-1H-6-20), following the manufacturer’s instructions. The absorbance was read at 450 nm wavelength using a microplate reader. The IL-6 and IL-1*β* quantification in the samples was calculated based on the standard curve.

### 4.13. Statistical Analysis

Linear mixed-effects models (LME) were fitted to the data.

For the measurements of FtH and Heme and the Western blotting (WB) of FPN, DMT1, TfR1, and p62, the following equation was used:Y = NF + Time + (1∣Sample)(1)

Two fixed effects were added: one for pathology (NF) and one for the experimental time point; for these experiments we had three temporal points (Time). A random effect for the sample (two patients and two controls) was also included.

For the analysis of morphology (Major/Minor Axis and Area), culture medium measurements (IL-6, IL-1β, and glutamate), and the WBs of NRF2, LCN2, EEAT2, MDA, and GPX4, as well as iron measurements (cytosolic LIP, mitochondrial LIP, and total iron) and SA-*β*-gal activity, the following equation used was:Y = NF + Day + NF∗Day + (1∣Sample)(2)

Here, the fixed effect “Time” was replaced with “Day” (representing the in vitro days 35 and 60 when data were collected). The interaction term NF∗Day was included.

For qRT-PCR data, the following equation was applied:Y = NF + Day + NF∗Day + (1∣Sample) + (1∣Session)(3)

An additional random effect for “Session” was included, representing mRNA extraction and the experimental plate used.

Fixed effects were evaluated using Type III ANOVA. Normality of dependent variable distributions and LME residuals was assessed graphically with QQ plots and histograms. All analyses were conducted in R (Version 4.4.0; https://www.r-project.org/, accessed 1 May 2024) using custom scripts.

## Figures and Tables

**Figure 1 ijms-26-06197-f001:**
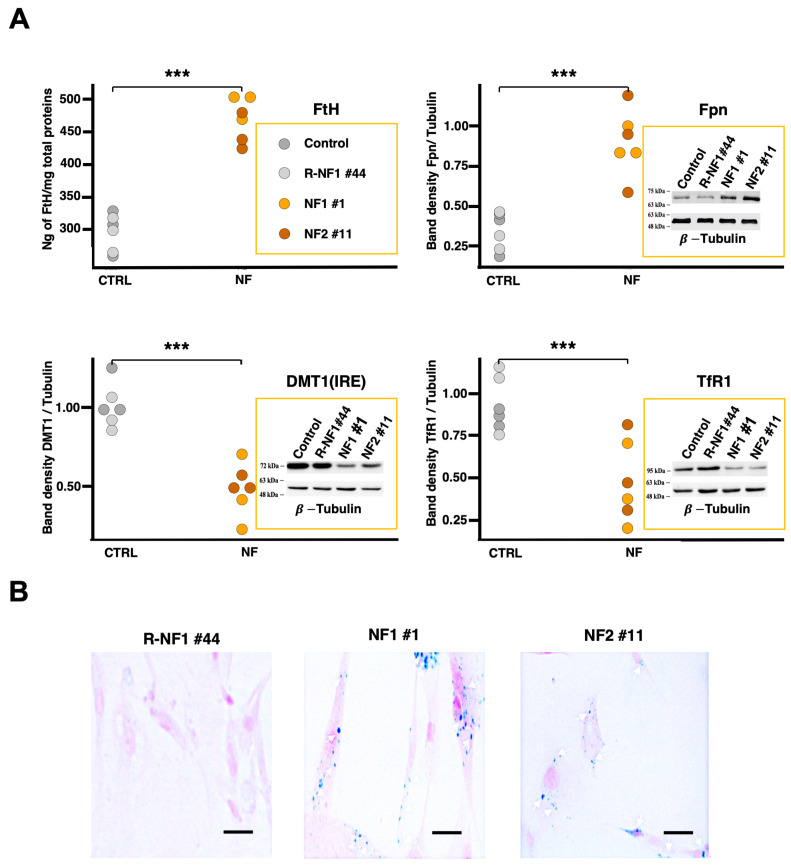
Iron dyshomeostasis in neuroferritinopathy. (**A**) Quantification of iron-related protein levels in differentiated astrocytes derived from hiPSCs. The top left panel displays the results of an ELISA assay measuring FtH levels. The top right, bottom left, and bottom right panels show the quantification of band densities from Western blot analyses of Fpn, DMT1, and TfR1, respectively. Experiments were conducted on cell lysates collected at different time points in vitro (days 35 and 60 of differentiation). The graphs represent data analyzed using linear mixed-effects (LME) models as described in the Materials and Methods section (Equation (1): Y = NF + Time + (1|Sample)), which included pathology (NF) as a fixed effect. The significance bracket above the groups indicates the overall statistical significance of the pathology effect across the tested time points, as determined by Type III ANOVA on the LME model (FtH: χ^2^_1_ = 120.1871, *p* < 0.0001; Fpn: χ^2^_1_ = 51.6462, *p* < 0.0001; DMT1: χ^2^_1_ = 69.6683, *p* < 0.0001; TfR1: χ^2^_1_ = 21.3097, *p* < 0.0001). (**B**) Representative images of Perls staining performed on d-astrocytes from neuroferritinopathy patients and healthy controls at day 60 of in vitro differentiation. The blue staining indicates the presence of ferric iron accumulation, which is visibly increased in patient-derived astrocytes compared to controls. Scale bars, 20 µm. *** *p* < 0.001, as determined by the LME model.

**Figure 2 ijms-26-06197-f002:**
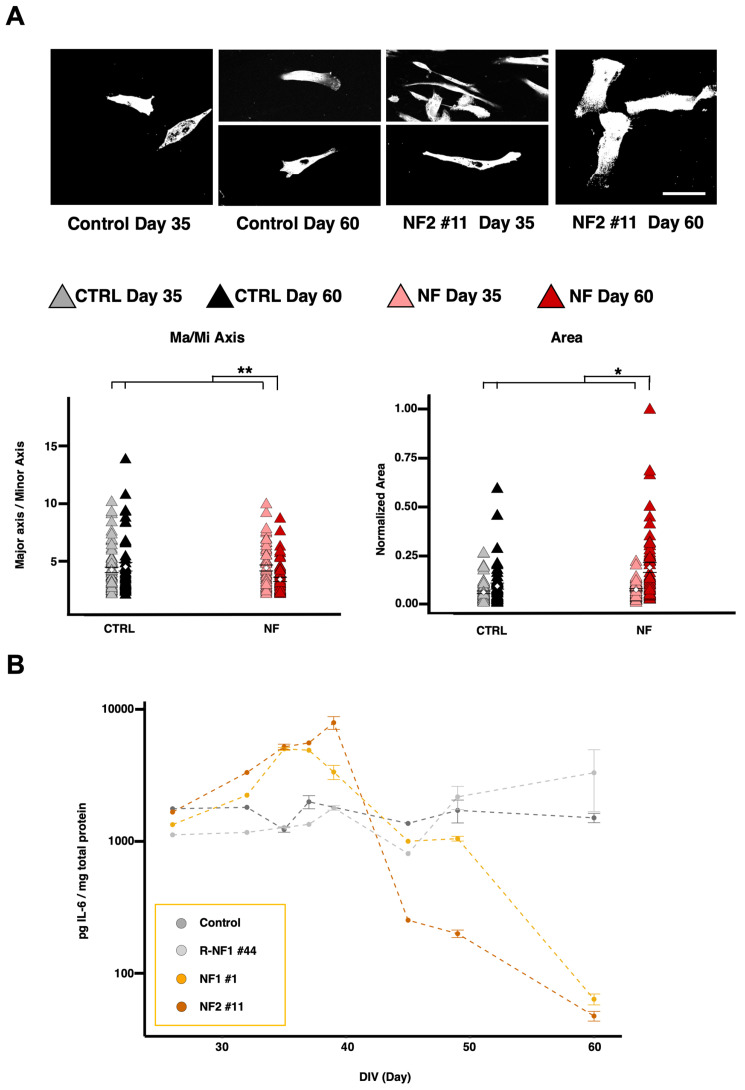
Astrocytic reactivity. (**A**) Morphological analysis of paraformaldehyde-fixed d-astrocytes immunostained with anti β-actin. Scale bars, 20 µm. The left graph displays the mean values with SEM for the major-to-minor axis ratio, while the right graph shows the mean values with SEM for the total area of the astrocytes. Individual data points are color-coded to represent different conditions: gray points indicate control astrocytes at day 35, black points represent control astrocytes at day 60, light red points represent patient-derived astrocytes at day 35, and dark red points represent patient-derived astrocytes at day 60. The graphs show results from two independent experiments analyzed using LME models as described in the Materials and Methods section (Equation (2): Y = NF + Day + NF∗Day + (1|Sample)), which included pathology (NF) as fixed effect, along with their interaction (NF × Day). Two significance brackets are displayed above each graph: the upper bracket indicates the statistical significance of the interaction effect between pathology and Day, whereas the lower bracket corresponds to the main effect of pathology across both time points (NF × Day for Major/Minor Axis: χ^2^_1_ = 7.8444, *p* = 0.0051; NF×Day for Area: χ^2^_1_ = 4.8357, *p* = 0.0279). (**B**) Graph showing the quantification of extracellular Interleukin-6 (IL-6) levels over time in the cell culture medium of differentiated astrocytes. The graph displays the mean values with SEM from two independent experiments. * *p* < 0.05, ** *p* < 0.01, as determined by the LME model.

**Figure 3 ijms-26-06197-f003:**
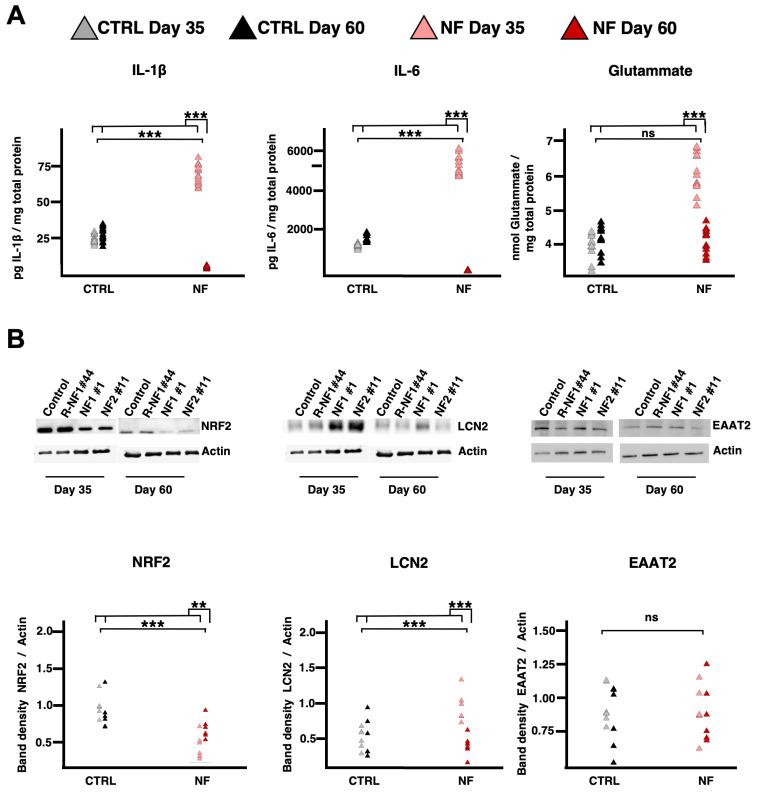
Astrocytic reactivity: extracellular composition analysis and Western blot. (**A**) Amount of IL-1β (**left**), IL-6 (**middle**), and glutamate (**right**) released by d-astrocytes normalized to cell total protein content. The graphs show the results of three independent experiments analyzed using LME models as described in the Materials and Methods section (Equation (2): Y = NF + Day + NF∗Day + (1|Sample)), which included pathology (NF) as fixed effect, along with their interaction (NF × Day). Two significance brackets are displayed above each graph: the upper bracket indicates the statistical significance of the interaction effect between pathology and day, whereas the lower bracket corresponds to the main effect of pathology across both time points (IL-1β = NF: χ^2^_1_ = 273.3277, *p* < 0.0001; Day: χ^2^_1_ = 9.5594, *p* = 0.0019; NF × Day: χ^2^_1_ = 1436.2533, *p* < 0.0001; IL-6 = NF: χ^2^_1_ = 1515.479, *p* < 0.0001; Day: χ^2^_1_ = 11.438, *p* = 0.0007; NF × Day: χ^2^_1_ = 1479.714, *p* < 0.0001). (**B**) Representative Western blot images (**top**) and related quantification graphs (**bottom**) of NRF2 (**left**), LCN2 (**middle**), and EAAT2 (**right**). Individual data points on the graphs are color-coded to represent different conditions: gray points indicate control astrocytes at day 35, black points represent control astrocytes at day 60, light red points represent patient-derived astrocytes at day 35, and dark red points represent patient-derived astrocytes at day 60. The graphs show results from three independent experiments analyzed using LME models as described in the Materials and Methods section (Equation (2): Y = NF + Day + NF∗Day + (1|Sample)), which included pathology (NF) as fixed effect, along with their interaction (NF × Day). Two significance brackets are displayed above each graph: the upper bracket indicates the statistical significance of the interaction effect between pathology and day, whereas the lower bracket corresponds to the main effect of pathology across both time points (NRF2: NF: χ^2^_1_ = 26.6379, *p* < 0.0001; NF × Day: χ^2^_1_ = 9.0012, *p* = 0.0027; LCN2: NF: χ^2^_1_ = 25.4127, *p* < 0.0001; NF × Day: χ^2^_1_ = 14.2569, *p* = 0.0002). *** *p* < 0.001, ** *p* < 0.01, as determined by the LME model.

**Figure 4 ijms-26-06197-f004:**
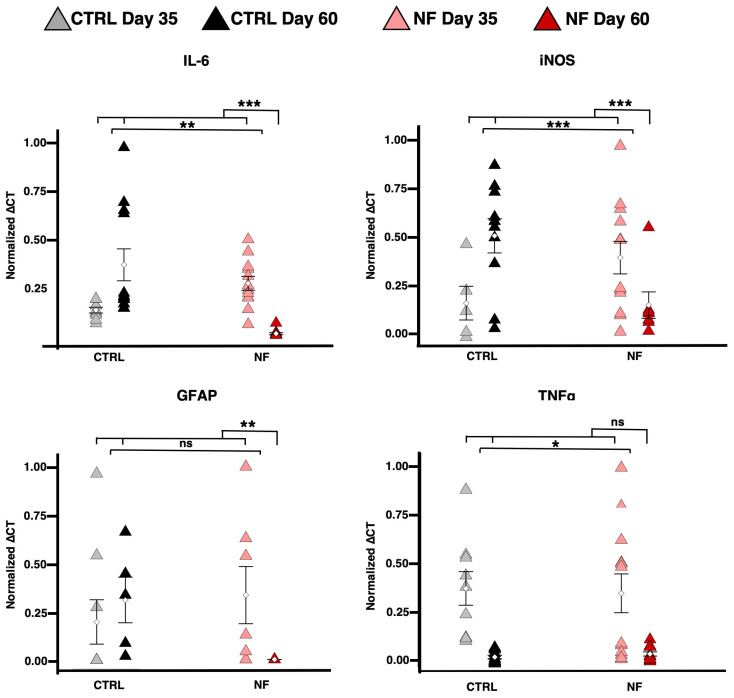
Astrocytic reactivity: rt-qPCR. Relative amount of IL-6 (top left, NF: χ^2^_1_ = 9.4851, *p* = 0.0021; Day: χ^2^_1_ = 7.2863, *p* = 0.0069; NF × Day: χ^2^_1_ = 90.1495, *p* < 0.0001), iNOS (top right, NF: χ^2^_1_ = 20.6667, *p* < 0.0001; Day: χ^2^_1_ = 13.2794, *p* = 0.0003; NF × Day: χ^2^_1_ = 47.0830, *p* < 0.0001), GFAP (bottom left, NF×Day: χ^2^_1_ = 8.6265, *p* = 0.0033), and TNFα gene (bottom right) expression in d-astrocytes tested by RT-qPCR. The graphs display the mean values with SEM from three independent experiments analyzed using LME models as described in the Materials and Methods section (Equation (3): Y = NF + Day + NF∗Day + (1|Sample) + (1|Session)), which included pathology (NF) as fixed effect, along with their interaction (NF × Day). Two significance brackets are displayed above each graph: the upper bracket indicates the statistical significance of the interaction effect between pathology and day, whereas the lower bracket corresponds to the main effect of pathology across both time points. * *p* < 0.05, ** *p* < 0.01, *** *p* < 0.001, as determined by the LME model.

**Figure 5 ijms-26-06197-f005:**
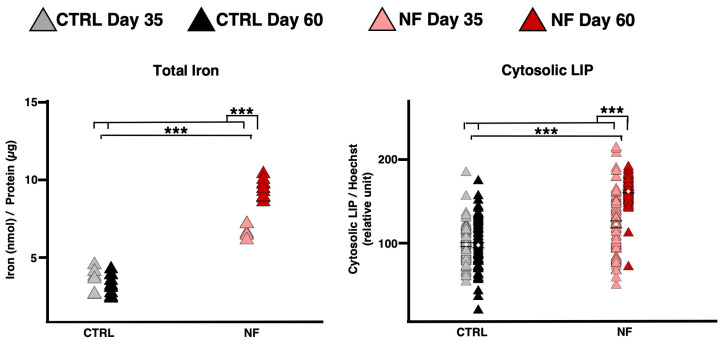
Iron variation over two time intervals. Graph showing the amount of total iron in astrocytes normalized to total protein concentration on days 35 and 60 (**left**) from three independent experiments; evaluation of cytosolic LIP on days 35 and 60 in d-astrocytes stained with the specific probe calcein (**right**) from two independent experiments, displaying the mean values with SEM. Two significance brackets are displayed above each graph: the upper one represents the statistical significance of the interaction effect, whereas the lower one corresponds to the main effect of pathology. (Total Iron: NF: χ^2^_1_ = 82.9721, *p* < 0.0001; NF × Day: χ^2^_1_ = 58.1272, *p* < 0.0001; Cytosolic LIP: NF: χ^2^_1_ = 36.2617, *p* < 0.0001; NF × Day: χ^2^_1_ = 43.4074, *p* < 0.0001). *** *p* < 0.001, as determined by the LME model.

**Figure 6 ijms-26-06197-f006:**
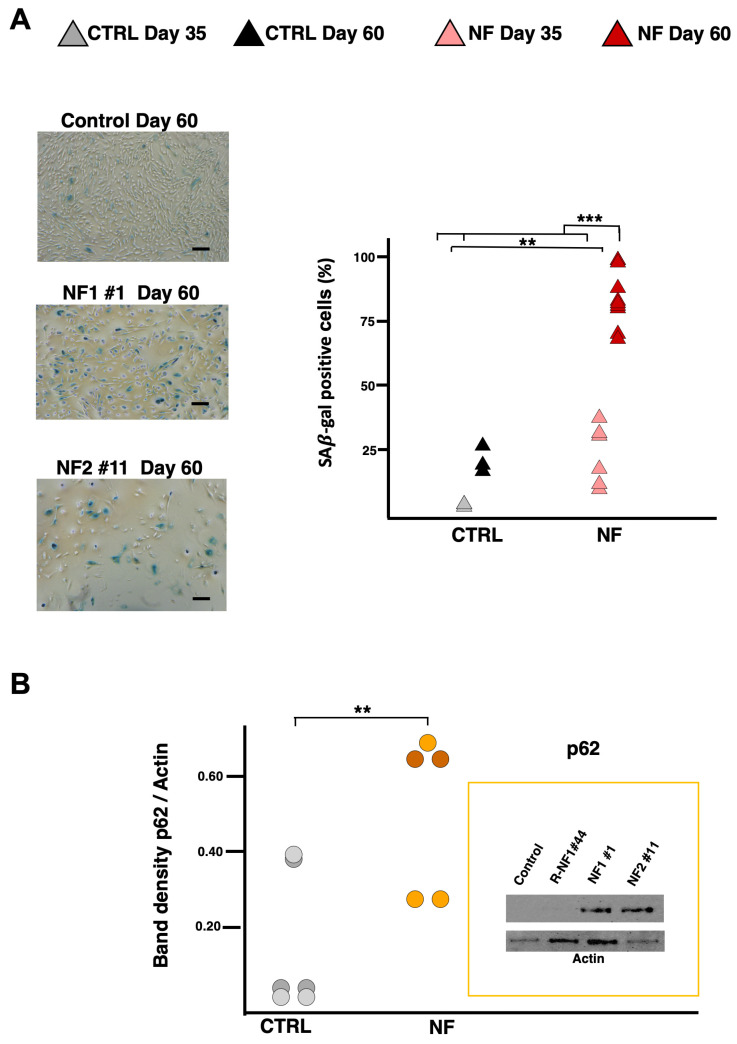
Cellular senescence. (**A**) Representative bright field images of SA-β-gal activity staining in control and NF patient-derived d-astrocytes. Scale bars, 20 µm. Positively stained (blue) cells were counted and plotted as the percentage of the total cells. Individual data points are color-coded to represent different conditions: gray points indicate control astrocytes at day 35, black points represent control astrocytes at day 60, light red points represent patient-derived astrocytes at day 35, and dark red points represent patient-derived astrocytes at day 60. The graph shows results from three independent experiments analyzed using LME models as described in the Materials and Methods section (Equation (2): Y = NF + Day + NF∗Day + (1|Sample)), which included pathology (NF) as fixed effect, along with their interaction (NF × Day). Two significance brackets are displayed above the graph: the upper bracket indicates the statistical significance of the interaction effect between pathology and day, whereas the lower bracket corresponds to the main effect of pathology across both time points (NF: χ^2^_1_ = 7.3874, *p* = 0.006568; DAY: χ^2^_1_ = 10.0753, *p* = 0.001503; NF × DAY: χ^2^_1_ = 44.1732, *p* < 0.0001). (**B**) Representative Western blot image of soluble cell homogenates from differentiated astrocytes at day 60, probed with anti-p62 antibody and anti-Actin antibody (as a loading control), with a graph showing the quantified protein levels via densitometry. The graph displays results from three independent experiments analyzed using LME models as described in the Materials and Methods section (Equation (1): Y = NF + Time + (1|Sample)), which included pathology (NF) as a fixed effect. The significance bracket above the groups indicates the overall statistical significance of the pathology effect (NF: χ^2^_1_ = 2.993, *p* = 0.0151). ** *p* < 0.01, *** *p* < 0.001, as determined by the LME model.

**Figure 7 ijms-26-06197-f007:**
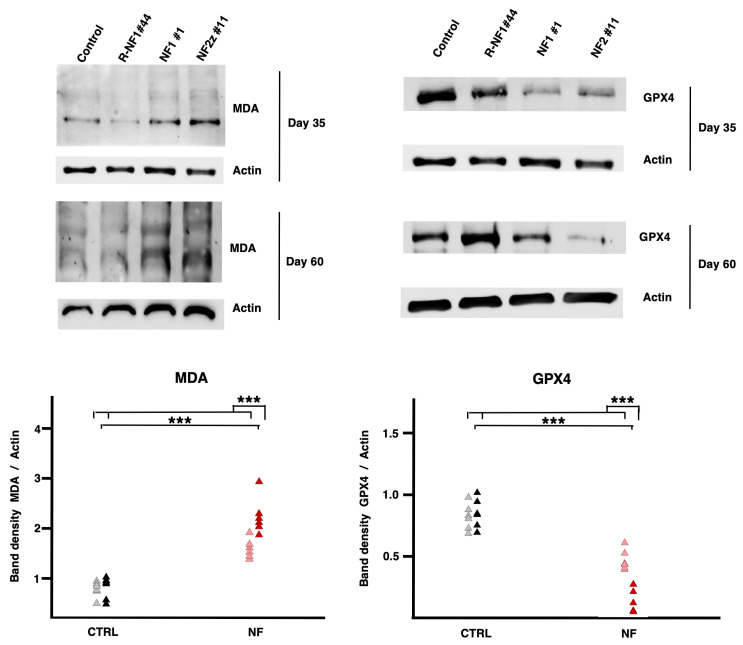
Ferroptosis over two time intervals. Representative Western blot images (**top**) and related quantification graphs (**bottom**) of MDA (left, NF: χ^2^_1_ = 31.7777, *p* < 0.0001; NF × Day: χ^2^_1_ = 12.6027, *p* = 0.0004) and GPX4 (right, NF: χ^2^_1_ = 24.0535, *p* < 0.0001; NF× Day: χ^2^_1_ = 26.5102, *p* < 0.0001), both measured at two time points (day 35 and day 60). Individual data points on the graphs are color-coded to represent different conditions: gray points indicate control astrocytes at day 35, black points represent control astrocytes at day 60, light red points represent patient-derived astrocytes at day 35, and dark red points represent patient-derived astrocytes at day 60. The graphs show results from three independent experiments analyzed using LME models as described in the Materials and Methods section (Equation (2): Y = NF + Day + NF∗Day + (1|Sample)), which included pathology (NF) as fixed effect, along with their interaction (NF × Day). Two significance brackets are displayed above each graph: the upper bracket indicates the statistical significance of the interaction effect between pathology and day, whereas the lower bracket corresponds to the main effect of pathology across both time points. *** *p* < 0.001, as determined by the LME model.

## Data Availability

The datasets generated and/or analyzed during the current study are available from the corresponding authors upon reasonable request.

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
