# Peer review of "Neuroferritinopathy Human-Induced Pluripotent Stem Cell-Derived Astrocytes Reveal an Active Role of Free Intracellular Iron in Astrocyte Reactivity"

_ijms, 2025, doi:10.3390/ijms26136197_

Round 1

Reviewer 1 Report

Comments and Suggestions for Authors

This manuscript presents a compelling investigation into the role of iron accumulation in astrocyte reactivity, with a particular focus on Neuroferritinopathy, a genetic neurodegenerative disorder characterized by primary iron overload. The authors employ hiPSC-derived astrocyte models to dissect the contribution of iron dysregulation to neuroinflammatory processes—an area of growing interest in the context of neurodegenerative diseases.

The study is well-motivated by the established observation that elevated brain iron levels are commonly associated with disease severity in various neurodegenerative conditions. The authors build on this by exploring how iron overload influences astrocytic behavior, a key element in the progression of neuroinflammation.

The use of patient-derived astrocytes from individuals with neuroferritinopathy represents a strength of the study, providing disease-relevant insights. The authors confirm iron accumulation and associated deregulation of iron-handling proteins in their model, and subsequently demonstrate that by Day 35 of differentiation, the astrocytes exhibit a reactive phenotype. This is convincingly supported by increased extracellular levels of IL-6, IL-1β, and glutamate, as well as morphological and molecular markers of reactivity and ferroptosis. A particularly interesting observation is the temporal shift observed by Day 60. Despite persistent iron overload, the pro-inflammatory cytokine levels decrease below those in control cells, and many reactivity markers show a reversal. However, increased senescence and continued ferroptotic activity at this later time point suggest a shift from an inflammatory to a degenerative cellular state.

Overall, the study provides novel evidence supporting the role of iron as a primary driver of astrocyte reactivity and underscores the importance of temporal dynamics in this process. The work contributes significantly to our understanding of iron’s role in neurodegeneration and offers a valuable model for future studies.

Major points:

  • Further clarification on whether the observed reversal at Day 60 is cell-intrinsic or reflective of a compensatory mechanism.
  • In the development of hiPS-derived astrocytic cultures, the characterization of the cellular phenotype is lacking, especially the control of the presence of other cells of glial nature.
  • Supplementary figures and tables are missing. Authors forgot to upload this data.
  • Introduction must be implemented with a mention on the role of microglia in the neuroinflammatory process. There is an imbalance in the genetic description as well as the inflammatory pathways.
  • Materials and methods have to be improved with more specific informations (i.e in the immunoblotting create a list of all the proteins analysed)
  • Discussion must be implemented with description of the potential therapeutic implications of modulating iron levels or ferroptosis in astrocytes. In general, in the discussion better resume the results described.
  • In Figure 1 add the blot images relative to the band density graphs
  • Legends need to be implemented and need to describe the figures represented more completely

Author Response

Referee #1

We thank the referee for her/his appreciation of our manuscript and for the very kind and helpful comments and suggestions.

Major points:

Comment:

Further clarification on whether the observed reversal at Day 60 is cell-intrinsic or reflective of a compensatory mechanism.

Response:

The reversal observed in patients at Day 60 probably reflects a compensatory mechanism triggered by iron overload, and not a cell-intrinsic effect, since control cells behave differently. We have previously demonstrated that in the d-astrocytes of other NBIA (Cozzi 2025), iron overloads leads to senescence. We added this concept in the discussion (lanes 292-293).

Comment:

In the development of hiPS-derived astrocytic cultures, the characterization of the cellular phenotype is lacking, especially the control of the presence of other cells of glial nature.

Response:
The characterization of the cellular model was present in the supplemental materials that we have loaded into the system but due to some technical problem they were not visible, we apologize for this. Now we have re-loaded the Supplemental material that report the characterization of the model. Nevertheless, our hiPSC-derived astrocytes were generated not only using the same protocol our laboratory has successfully applied in previous studies (Cozzi et al., 2025; Santambriogio et al., 2024) but also via a rigorously validated, literature-endorsed method that has been shown to yield an exceptionally pure astrocyte population, with minimal contamination by other glial cell types.

Comment:

Supplementary figures and tables are missing. Authors forgot to upload this data.

Response:
We apologize for this omission and have now re-uploaded the complete set of supplementary figures and tables to ensure full transparency of our data.

Comment:

Introduction must be implemented with a mention on the role of microglia in the neuroinflammatory process. There is an imbalance in the genetic description as well as the inflammatory pathways.

Response:
As recommended, we have expanded the final part of the Introduction to better address the role of neuroinflammation, with a specific mention of microglia and their involvement in the inflammatory response (lanes 118-128)

Comment:

Materials and methods have to be improved with more specific informations (i.e in the immunoblotting create a list of all the proteins analysed)

Response:

The table with the list of antibodies used (and diluition) was prepared and included in the Supplementary Materials but unfortunately was missing. We have now re-uploaded the table and we hope now is available

Comment:

Discussion must be implemented with description of the potential therapeutic implications of modulating iron levels or ferroptosis in astrocytes. In general, in the discussion better resume the results described.

Response:

As requested, we have expanded the final part of the Discussion to include a section on the potential therapeutic implications of modulating iron levels and ferroptosis in astrocytes. Additionally, we have revised the Discussion to better summarize and contextualize our main findings.

Comment:

In Figure 1 add the blot images relative to the band density graphs

Response:

We have updated Figure 1 by including an example of the blot images corresponding to the band density graphs.

Comment:

Legends need to be implemented and need to describe the figures represented more completely

Response:

We have revised and expanded the figure legends to provide a more complete and detailed description of the data presented.

Reviewer 2 Report

Comments and Suggestions for Authors

These authors analyzed the role of free intracellular iron in astrocyte reactivity using human induced pluripotent cells derived from a patient with a genetic neurodegenerative disorder neuroferritinopathy. This manuscript should be revised in terms of its style before its content. A portion of the comments are described below.

In the title and the abstract, the abbreviation of hiPS must be defined. It is not a general term.

I think a citation is needed in the first sentence of the introduction as below.

“The dysregulation of iron homeostasis in the brain is a common feature across many neurodegenerative diseases, with the precise role of iron in the progression of neurodegeneration still under debate. While the role of neuroinflammation in neurodegenerative processes is becoming increasingly well defined, the involvement of iron in neuroinflammation remains largely mysterious.”

Line 44: FLT1 must be italicized.

Line 66: “[10]–[18]of mutations…” should be mofidied appropriately.

Line 75: c.469_484dup mutation is not defined.

Line 79: The FTH gene was not defined.

Lines 96 and 97, how are FtL and FtH different? Not defined correctly.

Line 105: The position of Ref [28] is not correct, maybe.

Line 127: “351delG_InsTTT mutation” is not defined. I know that this is the correct way to write genetics, but in a manuscript, the authors need to explain and define exactly what the mutation is at the beginning of the abbreviation; for example, “The 351st guanine residue is deleted, and three thymine residues are inserted (351delG_InsTTT).”

The line breaks are in the wrong place (Section 2.3)

I cannot find Supplemental Figures.

In all figures, the sample size and the trial numbers are not represented in the legend. Furthermore, it is difficult to recognize which groups are compared in the significant differences.

Author Response

We thank the referee for her/his appreciation of our manuscript and for the very kind and helpful comments and suggestions.

Comment:

In the title and the abstract, the abbreviation of hiPS must be defined. It is not a general term.

Response:
We thank the reviewer for the remark. We have revised the title and the abstract to define the abbreviation hiPS, clarifying its meaning upon first mention.

Comment:

I think a citation is needed in the first sentence of the introduction as below.

“The dysregulation of iron homeostasis in the brain is a common feature across many neurodegenerative diseases, with the precise role of iron in the progression of neurodegeneration still under debate. While the role of neuroinflammation in neurodegenerative processes is becoming increasingly well defined, the involvement of iron in neuroinflammation remains largely mysterious.”

Response:
As recommended, we have added an appropriate citation to support the first sentence of the Introduction.

Comment:

Line 44: FLT1 must be italicized.

Response:

The gene name FLT1 has been italicized as requested.

Comment:

Line 66: “[10]–[18]of mutations…” should be modified appropriately.

Response:

The formatting of the citation in line 66 has been corrected accordingly.

Comment:

Line 75: c.469_484dup mutation is not defined.

Response:

We have added additional information regarding the c.469_484dup mutation in the Materials and Methods section of our study

Comment:

Line 79: The FTH gene was not defined.

Response:

We have corrected that section and defined the FTH gene accordingly.

Comment:

Lines 96 and 97, how are FtL and FtH different? Not defined correctly.

Response:

We have provided a clearer explanation in that section regarding the differences between FtL and FtH.

Comment:

Line 105: The position of Ref [28] is not correct, maybe.

Response:

We have checked, and the position of Ref seems correct to us.

Comment:

Line 127: “351delG_InsTTT mutation” is not defined. I know that this is the correct way to write genetics, but in a manuscript, the authors need to explain and define exactly what the mutation is at the beginning of the abbreviation; for example, “The 351st guanine residue is deleted, and three thymine residues are inserted (351delG_InsTTT).”

Response:

We have provided a clearer explanation of the 351delG_InsTTT mutation in the Materials and Methods section, adding the specific alteration in the protein .

Comment:

The line breaks are in the wrong place (Section 2.3)

Response:

We have reviewed Section 2.3 and were unable to identify the issue with the line breaks.

Comment:

I cannot find Supplemental Figures.

Response:

There was an issue with uploading the supplementary materials; we had re-upload them.

Comment:

In all figures, the sample size and the trial numbers are not represented in the legend.

Response:

We have revised all figure legends to include the sample size and trial numbers.

Comment:

Furthermore, it is difficult to recognize which groups are compared in the significant differences.

 Response:

Unfortunately, it is not straightforward to display statistical significance when using advanced models such as linear mixed models. However, we have followed the same style as in our recent publication (Cozzi et al., 2025). To improve clarity, we have added further details in the figure legends.

Round 2

Reviewer 1 Report

Comments and Suggestions for Authors The authors have addressed the reviewers' comments appropriately, and the manuscript has been significantly improved. The revisions are clear and well integrated into the text.   However, the supplementary data that were previously requested are still missing. These additional data are important to fully support the study's conclusions and to allow for a more comprehensive evaluation of the results.   I recommend that the authors provide the necessary supplementary materials before the manuscript can be considered for acceptance.

Author Response

Comment: The authors have addressed the reviewers' comments appropriately, and the manuscript has been significantly improved. The revisions are clear and well integrated into the text.   However, the supplementary data that were previously requested are still missing. These additional data are important to fully support the study's conclusions and to allow for a more comprehensive evaluation of the results.   I recommend that the authors provide the necessary supplementary materials before the manuscript can be considered for acceptance.

Response:

Dear Reviewer

I apologize for this, we have uploaded the supplemental material file into the system twice but in both cases there must have been a technical problem. Now I have tried to redo the pdf and upload it again, I hope that now it can be viewed. I also write to the publisher sending it directly, hoping that the problem will be solved, thanks for the positive evaluation
Best regards

Sonia Levi

Reviewer 2 Report

Comments and Suggestions for Authors

This manuscript is not well-written and not well-styled. So many issues that should be revised have still been included in this manuscript.

The comparables for which no differences are found are described as significantly different.

The figure legend still lacks explanation.

Consequently, this manuscript lacks critical issues.

Author Response

Comment 1: This manuscript is not well-written and not well-styled. So many issues that should be revised have still been included in this manuscript.

We  changed the text accordingly to all reviewers' suggestion in the first run revision. At this point we were not understanding where the reviewer 2 requires improvement.

Comment 2: The comparables for which no differences are found are described as significantly different.

We have cheked again the statistic and graphs without finding errors. May be this occurs by a misunderstanding of the type of the statistical analisys.

Linear Mixed-Effects (LME) models offer significant advantages over traditional statistical analyses like t-tests and ANOVA, providing greater flexibility and robustness when analyzing complex data. Traditional analyses often rely on assumptions of independence and homogeneity of variance, which are frequently violated in contemporary research, particularly in longitudinal or clustered studies. LME models overcome these limitations by effectively handling repeated measures and longitudinal data, modeling the correlation between observations from the same subject through random effects. They are also more effective in managing missing data compared to ANOVA, which often resorts to complete case deletion. LME models can also analyze hierarchical or clustered data, modeling variability at different levels, and offer greater flexibility in specifying covariance structures, allowing for a better reflection of dependencies in the data. Finally, LME models are also suitable for unbalanced experimental designs. Scientific literature supports the use of LME models across various fields, highlighting their benefits in addressing data dependency and managing complex datasets (Pinheiro, J. C., & Bates, D. M. (2000). Linear mixed-effects models: basic concepts and examples. Mixed-effects models in S and S-Plus, 3-56).

Despite the power of LME models in statistical analysis, the graphical representation of group differences, particularly in the presence of interaction effects, poses significant challenges. An interaction effect occurs when the effect of one independent variable (pathology) on the dependent (q-PCR, western blot quantification, etc.) variable relies on the level of another independent variable (time). Visualizing main effects is relatively straightforward, but incorporating interactions is complex due to the limitation of two-dimensional space. In conclusion, LME models offer notable benefits in the analysis of complex data compared to traditional methods. However, the visualization of group differences and interaction effects remains a challenge, requiring careful consideration of graphical techniques and their integration with robust statistical analyses to ensure accurate interpretation of research findings. For the graphical representation of significance, we drew inspiration from a previous work employing the same analysis (Cozzi et al., 2025; Moro, A. S.,  et al. (2023).

The figure legend still lacks explanation.

Thank you for the helpful suggestion, the figure leggends were expanded and improved.